# Neuron-Radial Glial Cell Communication via BMP/Id1 Signaling Is Key to Long-Term Maintenance of the Regenerative Capacity of the Adult Zebrafish Telencephalon

**DOI:** 10.3390/cells10102794

**Published:** 2021-10-19

**Authors:** Gaoqun Zhang, Luisa Lübke, Fushun Chen, Tanja Beil, Masanari Takamiya, Nicolas Diotel, Uwe Strähle, Sepand Rastegar

**Affiliations:** 1Institute of Biological and Chemical Systems-Biological Information Processing (IBCS-BIP), Karlsruhe Institute of Technology (KIT), Postfach 3640, 76021 Karlsruhe, Germany; gaoqun.zhang@kit.edu (G.Z.); luisa.luebke@kit.edu (L.L.); fushun.chen@kit.edu (F.C.); Tanja.Beil@kit.edu (T.B.); masanari.takamiya@kit.edu (M.T.); 2Diabète Athérothrombose Thérapies Réunion Océan Indien, INSERM, UMR 1188, Université de La Réunion, 97400 Saint-Denis de La Réunion, France; nicolas.diotel@univ-reunion.fr; 3Centre of Organismal Studies, University Heidelberg, Im Neuenheimer Feld 230, 69120 Heidelberg, Germany

**Keywords:** adult neurogenesis, regeneration, quiescence, telencephalon, radial glial cell, neural stem cell, BMP, *id1*, notch, *her4.1*, zebrafish

## Abstract

The central nervous system of adult zebrafish displays an extraordinary neurogenic and regenerative capacity. In the zebrafish adult brain, this regenerative capacity relies on neural stem cells (NSCs) and the careful management of the NSC pool. However, the mechanisms controlling NSC pool maintenance are not yet fully understood. Recently, Bone Morphogenetic Proteins (BMPs) and their downstream effector Id1 (Inhibitor of differentiation 1) were suggested to act as key players in NSC maintenance under constitutive and regenerative conditions. Here, we further investigated the role of BMP/Id1 signaling in these processes, using different genetic and pharmacological approaches. Our data show that BMPs are mainly expressed by neurons in the adult telencephalon, while *id1* is expressed in NSCs, suggesting a neuron-NSC communication via the BMP/Id1 signaling axis. Furthermore, manipulation of BMP signaling by conditionally inducing or repressing BMP signaling via heat-shock, lead to an increase or a decrease of *id1* expression in the NSCs, respectively. Induction of *id1* was followed by an increase in the number of quiescent NSCs, while knocking down *id1* expression caused an increase in NSC proliferation. In agreement, genetic ablation of *id1* function lead to increased proliferation of NSCs, followed by depletion of the stem cell pool with concomitant failure to heal injuries in repeatedly injured mutant telencephala. Moreover, pharmacological inhibition of BMP and Notch signaling suggests that the two signaling systems cooperate and converge onto the transcriptional regulator *her4.1*. Interestingly, brain injury lead to a depletion of NSCs in animals lacking BMP/Id1 signaling despite an intact Notch pathway. Taken together, our data demonstrate how neurons feedback on NSC proliferation and that BMP1/Id1 signaling acts as a safeguard of the NSC pool under regenerative conditions.

## 1. Introduction

Adult mammals have a limited ability to generate new neurons and to repair injured nervous tissue [1]. In contrast, the brain of adult zebrafish contains many distinct neurogenic niches [2,3,4], and also unlike mammals, zebrafish can repair large brain lesions efficiently, frequently recovering function without striking disabilities [2,5,6,7,8]. In the telencephalon of adult zebrafish, the entire ventricular zone produces new neurons under physiological conditions representing the baseline or constitutive neurogenesis [2,6,7,9,10,11]. When the telencephalon is injured, proliferation of NSCs is transiently boosted above the baseline of constitutive neurogenesis with a peak at five to seven days after injury [7,10,12,13]. Peculiarly, injury inflicted within one hemisphere of the telencephalon leads to a proliferative response of NSCs only in this injured hemisphere, while no response is observed in the closely apposed stem cell niche of the uninjured half of the telencephalon [7,14,15]. Thus, the signals which trigger stem cell proliferation in response to injury remain confined within the injured hemisphere.

The ventricular zone of the adult zebrafish telencephalon is densely populated by the cell bodies of radial glia cells (RGCs), the NSCs of the zebrafish telencephalon [11,16]. RGCs express typical NSC-markers such as Glial acidic fibrillary protein (Gfap), Brain lipid binding protein (Blbp), and the Calcium-binding protein β (S100β) [9,10,16,17,18]. Under homeostatic conditions, the majority of the NSCs are quiescent type I RGCs. Only a small proportion of NSCs proliferate and express proliferation markers, such as proliferating cell nuclear antigen (PCNA). This latter so-called type II RGCs can give rise to committed neuronal progenitors corresponding to neuroblasts (type III cells) [10,16]. When the telencephalon is injured, many more NSCs enter the cell cycle and start to express proliferation markers [7]. Concomitantly, the NSCs generate an increased number of new neurons compared to homeostatic conditions [19]. Newborn neuronal precursors migrate from the ventricular layer to the injury site to replace lost neurons [6]. This regenerative neurogenesis can be initiated by inflammatory signals [20,21,22].

During regeneration, NSCs were proposed to divide mostly symmetrically giving rise to two new neurons or two NSCs [19]. Especially, the increase in neurogenesis in response to injury could rapidly lead to depletion of stem cells unless if these fate decisions are not carefully managed. Genetic and pharmacological evidence strongly support a role of Notch signaling in regulating NSC quiescence, both in mouse and zebrafish [23,24,25,26]. Specifically, in the zebrafish telencephalon, Notch3 is expressed in RGCs and promotes quiescence of NSCs [27]. This signaling originates, at least in part, from the progenitor population itself [28,29]. Notch signaling appears to serve as an intrinsic signaling cue for the regulation of quiescence and the maintenance of the stem cell population in mammals and fish [1,9].

Previously, in a genome-wide search for transcription regulator genes differentially expressed in response to telencephalic injury in the zebrafish [14,30,31], we identified the HLH factor Id1 which is a non-DNA-binding inhibitor of basic-Helix-Loop-Helix (bHLH) transcription factors [32]. *id1* was shown to be mainly expressed in quiescent NSCs and to be specifically up-regulated in these cells after injury [14,15]. As in constitutive neurogenic condition, *id1* expression was mainly associated with non-proliferative NSCs after brain injury [14]. Functional studies using morpholinos and mosaic gain-of-function by in situ lipofection suggested a role of Id1 in repressing proliferation of NSCs. Therefore, we speculated that in particular its up-regulation is important to maintain the NSC pool after injury [14]. Id factors have previously been implicated to play a role in the maintenance of the stem cell pool in mice indicating a possible conservation of the mechanism [33,34,35]. However, mice do not exhibit the same strong regenerative capacity as zebrafish [9].

Further investigation revealed that expression of *id1* in NSCs is driven by a phylogenetically conserved *cis*-regulatory module (CRM) [15]. For activity under both constitutive and regenerative conditions, this CRM requires Smad1/5 and Smad4 binding motifs which are intracellular mediators of BMP signaling [15]. Our preliminary data suggested thus that, in addition to Notch, BMP signaling may play a role in regulating the proliferative activity of NSCs during constitutive and regenerative neurogenesis in the zebrafish telencephalon [15].

Given the degenerate nature of transcription factor binding sites and the promiscuity with which these are bound by TFs, our previous data are suggestive but not proving. Moreover, the concern regarding the unspecific effects of morpholinos demand repetition of previous experiments with more reliable methods. In recent years, it has been recognized that lipofection can induce inflammatory responses [36]. In light of the known influences of inflammation on cell proliferation in the zebrafish telencephalon, previous results have to be confirmed using alternative methods. We therefore embarked on complementary genetic gain- and loss-of-function experiments with stably inherited transgenes and mutations. Our approaches have the advantage, that we can address the so far unanswered question whether *id1* is required for maintenance of the regenerative capacity of the zebrafish telencephalon.

Here, we tested two hypotheses: (1) BMP/Id1 signaling represses proliferation of NSCs in the zebrafish telencephalon and (2) this repression is required to maintain stem cell pools and thus the regenerative capacity of the adult zebrafish telencephalon. By conditional genetic approaches, we show that BMP signaling controls proliferation, as well as *id1* expression, consistent with the notion that BMPs/Id1 are regulators of stem cell quiescence. Moreover, genetic ablation of *id1* function leads to an increased proliferation of NSCs and subsequent depletion of stem cell pools with concomitant failure to heal injuries in repeatedly injured mutant telencephala. Our data show that BMP signaling is key to the maintenance of NSC pools and the long-term capacity to regenerate wounds in the zebrafish telencephalon.

## 2. Results

### 2.1. BMP Proteins Are Expressed in Neurons of the Telencephalon

Previous data suggested that BMPs may control expression of *id1* and in this way regulate cell proliferation in the telencephalon [15]. We first addressed the question of whether *bmps* are expressed in the telencephalon of the adult zebrafish. We chose *bmp2a*, *bmp2b*, *bmp4*, *bmp7a* and *bmp7b*, all of which are expressed at low levels in the telencephalon [14,37] (and data not shown) and can act redundantly on the same pathways [38,39]. We carried out expression analyses on transverse sections through the telencephalon by in situ hybridization (ISH) with either chromogenic (Figure 1A–E) or fluorescent staining (Figure 1F–J) of antisense probes directed against mRNAs of *bmp* genes. Expression of the mRNAs of the five *bmp* genes was detected in the brain parenchyma and along the telencephalic periventricular layer in similar and overlapping patterns, although levels of expression varied between different sub-regions of the telencephalon for individual *bmp* genes (Figure 1A–E).

Next, we tested which cells express the *bmp* genes. To this end, we combined fluorescent in situ hybridization (FISH) for the five *bmp* probes with immunohistochemistry (IHC) using anti-HuC/D antibodies (Hu) to label post-mitotic neurons. By additional co-staining with antibodies directed against glutamine synthetase (GS), we marked NSCs in the telencephalic tissue sections. All five *bmp* genes were strongly expressed in neurons while their expression was not detectable in NSCs (Figure 1F–J). Thus, NSCs, with their cell bodies at the medial periventricular zones and their long processes traversing the parenchyma all the way to the pial surface, are embedded in a *bmp*-expressing, neuronal environment.

We next assessed whether this abundant expression of *bmp* mRNA in neurons also reflects the expression of the BMP protein. To address this, we conducted IHC using a BMP2b antibody (the only BMP antibody available for zebrafish) and an antibody directed against NeuroD1. To assess expression in NSCs, we carried out double IHC with the BMP2b antibody and the GS antibody marking NSCs. BMP2b was not expressed in NSCs (Figure 1K–K’’’, white arrowheads). However, and as expected from the ISH data, BMP2b expression was associated with the NeuroD1 expressing neurons (Figure 1L–L’’’, yellow arrowheads). We also checked whether oligodendrocytes marked by *Tg*(*olig2:gfp*) [40], co-express BMP2b. Only weak or no BMP2b staining was detected in oligodendrocytes (Appendix A).

These data are consistent with a model where neurons are BMP-producing and NSCs are BMP-sensing cells in the telencephalon. To assess whether NSCs are, in principle, able to respond to BMP signals, we mined single cell sequencing data [41] for the expression of components of the BMP signaling pathway (Appendix A). Quiescent NSCs were unequivocally identified by expression of *id1* and a lack of *cyclin D1* (*ccnd1*) transcripts [41]. All 22 cells identified with this expression pattern expressed the BMP receptors *bmpr1ab* and *bmpr2b* and the downstream BMP specific mediators *smad1*, *smad5* and *smad9* (Appendix A). Consequently, NSCs harbor the relevant components of the canonical BMP signaling pathway suggesting that they can respond to BMP signals. In addition, the BMP induced gene *bambia* is expressed in NSCs (Appendix A) in agreement with the notion that NSCs receive BMP signals.

### 2.2. Bmp Signaling Induces id1 Expression and Promotes Stem Cell Quiescence

To test the relevance of the observed BMP expression, we manipulated BMP signaling by conditionally inducing BMP signaling in adult zebrafish. As a BMP with representative activity of all the five expressed BMPS, we chose Bmp2b to trigger BMP expression with the help of a stably integrated, heat-shock inducible transgene *Tg*(*hsp70:bmp2b*) [42]. We checked the efficiency of the manipulation by employing an antibody directed against the phosphorylated states of Smad1, 5 and 9, the known intracellular mediators of the canonical BMP signaling cascade. Phosphorylation of these Smads reflects the activation of the serine/threonine kinase domains of the BMP receptors upon BMP ligand binding [43]. No phosphorylated Smad1/5/9 (pSmad) staining was detectable above background in GS-marked NSCs in the absence of heat-shock (Figure 2B–B’’). In striking contrast, when BMP2b was ectopically expressed by heat-shock, strong immunoreactivity of pSmads was noted in NSCs (Figure 2C–C’’). Wild-type (WT) animals not carrying the *Tg*(*hs:bmp2b*) transgene did not show pSmad staining upon heat-shock (Figure 2A–A’’). Thus, ectopic expression of Bmp2b leads to a strong activation of BMP signaling in NSCs. Of note, the induction of phosphorylation of Smad is restricted to the NSCs, suggesting that neurons do not respond to BMPs in a strong fashion under the employed induction scheme. This further supports our model that neurons are BMP sending and NSCs are BMP perceiving cells.

We next assessed whether BMP2b would affect *id1* expression. Indeed, an increase of *id1* mRNA was triggered by heat-shock in the ventricular zone in comparison to transgenic animals without heat-shock (Figure 2E,F and Appendix A). Heat-shock of a WT adult zebrafish did not elicit induction of *id1* (Figure 2D) demonstrating that the observed response was dependent on the transgene. Quantification of the experimental data revealed that the level of *id1* induction was in the range (Figure 2G) which was previously noted in response to injury [14].

We next asked whether the observed induction of *id1* is paralleled by a reduction of proliferating NSCs. For this purpose, sections through the telencephalon of heat-shocked and non-heat-shocked *Tg*(*hsp70:bmp2b*) animals were labeled with antibodies directed against the NSC marker S100β [16,44] and PCNA [16,45]. In comparison to non-heat-shock controls (Figure 2I,I’), the number of proliferating NSCs was reduced in brains of heat-shocked animals (Figure 2J,J’, White boxed area). Upon quantification, the reduction of proliferating NSCs was found to be highly significant (Figure 2K). Heat-shock of WT adult zebrafish did not elicit induction of NSC proliferation (Figure 2H,H’). Taken together, expression of Bmp2b leads to an increase of *id1* expression and a decrease of proliferating NSCs, in line with the notion that BMPs control cell proliferation via activation of *id1*. This pattern of decrease in proliferation and increase of *id1* expression is highly similar to that seen previously as a response to injury of the telencephalon [14]. 

### 2.3. Inhibition of BMP Signaling Causes Increased Proliferation of NSCs

If BMPs were indeed involved in the regulation of NSC proliferation, we would anticipate that blocking BMP signaling decreases *id1* expression and increases proliferation. To test this, we employed the conditional expression of a dominant-negative BMP receptor 1a mutant [46] in which the kinase domain has been deleted. In comparison to non-heat-shocked *Tg(hs:dnBmpr1a*) animals (Figure 3A), a heat-shock resulted in a decrease of *id1* expression in the telencephalon (Figure 3B). Meanwhile, heat-shock of WT animals not carrying the transgene did not affect *id1* expression (Appendix A). When quantified with real-time quantitative PCR (RT-qPCR), the induced reduction was confirmed to be highly significant (Figure 3C).

We next asked whether this induced reduction of *id1* expression is accompanied by an increase in NSCs proliferation. In comparison to non-heat-shocked controls (Figure 3D,D’), the *hs:dnBmpr1a* expressing animals showed an increase of PCNA+/S100β+ (white boxed area) and thus proliferating NSCs (Figure 3E,E’,F). These data demonstrate that inhibition of BMP signaling by employment of the dominant-negative receptor, leads to the reverse effect of what was observed upon forced expression of BMP2b. The proliferation of NSCs in the zebrafish telencephalon is thus controlled by BMPs. In conclusion, the presented data show that BMPs regulate *id1* expression and suppress cell proliferation.

### 2.4. Genetic Removal of id1 Activity Leads to Increased Proliferation in the Telencephalon

Previously, transient reduction of Id1 activity by cerebroventricular microinjection of a vivo-morpholino led to increased proliferation of NSCs [14]. Given the temporal and spatial limitation of this method and its potential toxic impact on the transfected tissue [47], we first sought to confirm the results from these transient studies with a stable genetic approach. We generated a knock-out of *id1* using a CRISPR-Cas9 approach. The identified mutant contains a 2 bp deletion in the start codon of the *id1* coding sequence, which leads to a non-functional shorter Id1 protein (Appendix A). In this mutant the level of *id1* mRNA expression was not notably affected (Appendix A). Homozygous *id1^ka706/ka706^* animals were viable and reached adulthood without gross morphological defects.

To assess the proposed role of *id1* in the regulation of proliferation of NSCs, we stained telencephalic cross-sections of 6-month-old WT (Figure 4A–A’’) and *id1^ka706/ka706^* animals (Figure 4B–B’’) with the NSC marker S100β and the proliferation marker PCNA. A two-fold increase in the number of PCNA+ cells (yellow arrows, inside white boxed area) could be detected in the ventricular zone of the mutants in comparison to that of WT siblings (Figure 4C). When the number of PCNA+/S100β+ cells (white arrows, inside white boxed area) was compared between mutant and WT, a similar increase was noted (Figure 4D), suggesting that in *the id1^ka706/ka706^* mutant about twice as many NSCs are in a proliferative state. These data confirm our preliminary results from vivo-morpholino knock-down experiments [14].

We next tested whether the increased number of proliferating NSCs has an effect on *ascl1a* expression which has been implicated in proliferation as well as commitment to neurogenesis in mammals [33,48,49]. To this end, we stained transverse sections through WT and *id1^ka706/ka706^* telencephala with an antisense probe directed against the neural precursor gene *ascl1a* (Figure 4E,F). A 1.5-fold increase of *ascl1a* expression was observed in the mutant telencephalon (Figure 4G) which was further verified and confirmed by RT-qPCR (Figure 4H).

Next, we asked whether the higher number of activated NSCs in the mutant would lead to an increase in the number of NSCs in the mutant. We found a slightly but significantly increased number of NSCs in the mutant telencephala (Appendix A). Thus, the pool of stem cells is not depleted despite the increased neurogenesis in the *id1* mutant. This suggests that mechanisms remain in operation that maintain the stem cell pool in the mutant under normal physiological conditions.

### 2.5. id1 Is Essential to Preserve Long Term Maintenance of the Regenerative Capacity in the Injured Telencephalon

The level of *id1* expression in individual cells, as well as the total number of cells, is increased five days after inflicting a lesion to the telencephalon. This increased response of *id1* expression followed the induction of NSC proliferation [14]. We therefore speculated that Id1 is required to dampen the proliferative response to prevent exhaustion of the stem cell pool in the regenerating brain. We addressed this hypothesis by causing repeated stab wounds in the telencephalon of *id1^ka706/ka706^* and WT control animals. In our wounding protocol, the right hemisphere of the telencephalon is injured by introducing a needle through the skull while the left hemisphere remains uninjured [7,50]. Afterwards, the animals were allowed to recover 1 month before a second lesion was inflicted (Figure 5A). Stab wounded animals were sacrificed 5 days after the first or the second lesion or kept for another 3 months (Figure 5A). For examination of proliferation and repair of lesion, sections through the telencephalon were co-stained with antibodies against the proliferation marker PCNA and the NSC marker S100β. As expected from the previous experiments, mutant animals showed a stronger proliferative response of the NSCs than the WT control, 5 days after the first lesion (compare Figure 5C,C’ with Figure 5B,B’, white boxed area). This is in striking contrast to the situation after the second injury: While WT siblings had many PCNA+ NSCs and thus a profound proliferative response to the second lesion, the mutants displayed less proliferating NSCs (Figure 5E–G). Moreover, when examined 3 months after the second wounding, the WT animals (*n* = 16) had all healed the second wound (Figure 5H,I) and only 5 animals displayed some mild traces of the injury (Figure 5I). In contrast, 7 of the 11 mutant animals presented massive tissue defects (Figure 5J) and only 4 had repaired the second lesion partially with clear signs of the lesion remaining (Figure 5K, white arrows). Also, the survival rate was slightly less in the mutants at 3 months (Appendix A).

We next asked whether the overall number of NSCs in the telencephalon decreased in the mutant. The number of NSCs (S100β+ RGCs) was comparable 5 days after the first lesion between the mutant and WT telencephala (Figure 5L). Note that uninjured animals showed a slightly increased number of NSCs (Appendix A). After the second injury, the number of NSCs (S100β+ RGCs) had dropped by 32% in the mutants (Figure 5M). Hence, lack of *id1* function causes not only loss of proliferation of NSCs but also shrinkage of the NSC pool as a whole in the injured animal.

In summary, in absence of *id1*, the ventricular zone initially displayed a much more profound proliferative response to a lesion. Upon repeated injury, however, proliferating NSCs were severely reduced in the mutant. This, together with the observed decrease of NSCs, demonstrates depletion of proliferation-competent stem cells. These data show that *id1* is required to maintain the stem cell pool after wounding and thus the long-term capacity to efficiently repair wounds in the telencephalon.

### 2.6. BMP and Notch Signaling Act Mostly in the Same NSCs

Notch3 and the Notch-target gene *her4.1* are mainly expressed in quiescent NSCs [27,51] and Notch3 negatively controls the proliferation of NSCs [27]. It was previously suggested that there are distinct classes of NSC populations with different proliferative potential [16,17,41,51,52,53]. Our observation that BMPs mediate NSC quiescence raises the question of whether these two pathways cooperate or whether they act on different stem cell populations.

In order to investigate this question, we first assessed whether *id1* and *notch3* or *her4.1* are expressed in the same NSCs (Figure 6). We hybridized sections through the telencephalon of *Tg*(*id1-CRM2:gfp*) fish with a *notch3* antisense RNA probe and counted cells expressing *notch3* and co-expressing *notch3* and the id1-CRM2 transgene (Figure 6A–C). In total, co-expression was detected in 83.7% of cells, suggesting that the vast majority of the NSC population is targeted by both, Notch3 and BMP signaling (Figure 6D). To confirm this, we crossed *Tg*(*her4.1:gfp*) transgenic fish [54] into a *Tg(id1-CRM2:mCherry)* background [15]. We counted 88.7% cells co-expressing the two transgenes (Figure 6E–H). This suggests that the two regulators of stem cell quiescence are largely co-expressed and therefore act in most cases in the same cells.

Curiously, we found significant, but much smaller fractions of NSCs that expressed only *Tg*(*her4.1:gfp*) or *Tg*(*id1-CRM2:mCherry*) (Appendix A–B’’’). We assessed whether there are differences in proliferative potential between the *her4.1+/id1+* NSCs and the NSCs expressing only one of the two genes. Indeed, 31.7% NSCs expressing *Tg*(*her4.1:gfp*) and 56.7% NSCs expressing *Tg*(*id1-CRM2:mCherry*) only were PCNA+, while only 4.7% NSCs positive for both markers were also PCNA+ (Appendix A). This suggests that cells expressing either *id1* or *her4.1* are more frequently cycling than the double positive cells. This observation is in agreement with the notion that the two pathways cooperate in NSCs to control quiescence.

### 2.7. BMP/id1 Signaling Affects Expression of her4.1

Despite the obvious relationship between BMP/Id1 signaling and Notch signaling, *id1* expression in the adult telencephalon is not affected by the Notch inhibitor LY (Ref. [14] and Appendix A)) suggesting that *id1* expression and hence also BMP signaling is independent of Notch signaling. Therefore, we examined the relationship from the opposite side and asked what happens to the expression of *her4.1*, if we block BMP signaling. Treatment with 10 μD an inhibitor of the canonical BMP signaling pathway [55], affected the expression of *her4.1* marginally as shown by ISH (Figure 7B) and confirmed by RT-qPCR (Figure 7F, second column from left). Application of 6 μM LY411575 (LY), a gamma-secretase inhibitor and potent Notch signaling blocker [56], strongly reduced the expression of endogenous *her4.1* entirely (Figure 7C,F (middle column)), while 3 μM LY (Figure 7D,F (second column from the right)) caused a reduction of *her4.1* expression relative to DMSO treated controls (Figure 7A,F (first column from the left). Co-application of 10 μM BMP inhibitor DMH1 together with 3 μM Notch inhibitor LY very strongly abolished *her4.1* expression (Figure 7E,F, first column from the right)). Thus, BMP signaling has, like Notch signaling, a positive effect on *her4.1* expression suggesting that the two pathways converge on this transcription factor.

Next, we asked whether cell proliferation is also affected by chemically blocking the Notch and BMP pathways. We co-stained telencephalic sections of inhibitor-treated animals with antibodies against PCNA and S100β (Figure 7A’–E’’). While exposure to 10 μM DMH1 alone had no effect on the number of PCNA + RGCs (Figure 7B’,B’’,G, second column from the left) addition of 6 μM LY (Figure 7C’,C’’,G, middle column) or 3 μM LY (Figure 7D’,D’’,G, second column from the right) caused an increase in proliferating cells. The rise in proliferation was more important with the addition of 6 μM LY (Figure 7G, middle column) than with 3 μM LY (Figure 7G, second column from the left). When suboptimal inhibitor concentrations (3 μM LY, 10 μM DHM1) were co-applied (Figure 7E), an even more notable increase in the number of proliferating cells was scored (Figure 7E’,E’’,G, first column from the right). This increase of proliferating NSCs in co-inhibited animals is similar to the effect on *her4.1* expression. These observations further support the notion that BMP and Notch signaling pathways converge on *her4.1*.

Inhibitor experiments are limited by the efficacy and specificity of the inhibitors. To confirm the suggested effect of BMP signaling on *her4.1* expression, we analyzed *her4.1* expression in transgenic *Tg*(*hs:dnBmpr1a*) animals expressing the dominant negative BMP receptor in response to heat-shock. When exposed to elevated temperature, *her4.1* expression was reduced in comparison to transgenic animals not exposed to heat-shock (Appendix A). Thus, *her4.1* appears to be under the control of BMP signaling. We also analyzed *her4.1* expression in *id1^ka706/ka706^* mutant animals. Transverse sections through WT and *id1^ka706/ka706^* telencephala stained with an antisense probe directed against the *her4.1* gene (Appendix A) show a reduction of *her4.1* expression in the *id1^ka706/ka706^* mutant compared to WT animals (Appendix A). This reduction was validated by RT-qPCR (Appendix A).

## 3. Discussion

Zebrafish have the remarkable ability to heal injuries in the adult central nervous system. Here, we show that, in the telencephalon, the basis of this long-term neurogenic potential involves the management of the stem cell pool by the BMP/Id1 signaling pathway. Interference with BMP signaling led to modulation of *id1* expression and proliferation of NSCs. Genetic ablation of *id1* activity caused increased proliferation of NSCs. Additionally, repeated wounding resulted in depletion of the stem cell pool and failure to repair the injury of the telencephalon. Our data are consistent with a model in which the regenerating nervous tissue feeds back on the stem cell pool via BMP/Id1 signaling thereby promoting quiescence after an initial rise of proliferation in response to injury. Cross-talk of BMP/Id1 signaling with the Notch signaling pathway appears to occur at the level of expression of the transcription factor *her4.1*. In the absence of *id1* and under homeostatic conditions of constitutive neurogenesis, Notch signaling appeared to be sufficient to maintain the stem cell pool. However, when challenged by the need to repair multiple injuries, this safeguard is not sufficient anymore and the NSC pool is depleted in an *id1* mutant. Thus, the NSC pool is maintained by an at least two-tiered regulatory mechanism involving both Notch and BMP/Id1 signaling.

### 3.1. Neuron-Derived BMPs Control Proliferation of NSCs

Several lines of evidence demonstrate that BMPs control proliferation of NSCs. Conditional expression of BMP2b caused reduction of NSC proliferation while blocking the expression of BMP signaling via expression of dnBMPR1a led to increased proliferation of NSCs.

The source of BMPs in vicinity to the NSCs is the neurons in the telencephalic parenchyma. In contrast to stem cells in the mouse telencephalon [57], NSCs of the zebrafish telencephalon do not express BMPs. However, single cell-sequencing analysis suggests that NSCs are fully competent to respond to BMP signals expressing all the key members of the intracellular BMP signaling cascade. Moreover, when BMPs are over-expressed, only NSCs show detectable pSmad staining in cross-sections through the parenchyma of the telencephalon. These data suggest that neurons are the BMP emitting cells and NSCs are the BMP receiving cells.

*id1*, whose expression is increased by wounding [14], is known to depend on BMP signaling. *id1* expression in NSCs including its up-regulation in response to injury is faithfully mimicked by a cis-regulatory module derived from the id1 gene. This module requires putative Smad1/5/9 DNA binding elements to function [15]. By showing that BMP signaling is necessary to induce *id1* expression our functional data further support the notion that *id1* is a primary target of the BMP signaling cascade.

A noteworthy phenomenon is the restricted activation of proliferation in the injured hemisphere only [7,14]. Likewise, activation of *id1* expression occurs only in the injured hemisphere [14,15]. This suggests that not only induction of proliferation by injury but also its dampening by BMPs back to the baseline levels of constitutive neurogenesis is restricted to the injured hemisphere. It is thus unlikely that BMPs are released into the cerebrospinal fluid as they would otherwise also affect NSCs in the uninjured hemisphere. The long processes of NSCs spanning the entire parenchyma may play a role as sensors of BMP signals. It remains to be determined whether BMP expression is increased in response to repair of the tissue relative to normal levels of expression. By both RNA ISH and antibody staining, we did not detect changes of expression of the four tested *bmp* genes or the BMP2b protein (unpublished). However, some of the tested BMPs showed slightly elevated levels in comparative RNA-Seq of control and injured telencephalic hemispheres (unpublished). This weak response may be a matter of sensitivity or BMPs are regulated at the level of processing or other post-translational mechanisms that escaped our detection due to lack of tools. Irrespectively, our data suggest that neurons are the BMP producing and NSCs are the BMP sensing cells. This cross-talk is somehow up-regulated by regeneration leading to dampening of proliferation. Our data are consistent with a model in which the nervous system maintains, via BMP signaling, a feed-back loop between the regenerating nervous tissue and the stem cell niche thereby adapting the proliferative rate of the stem cells to the needs of tissue homeostasis and regeneration.

### 3.2. Id1 Mediates Long-Term Maintenance of the Regenerative Capacity

Mutation of *id1* leads to an increased number of proliferating NSCs and elevated formation of neuroblasts committed to a neural fate. These changes are similar in magnitude to those observed previously in response to injury of the telencephalon [14,15]. Mechanisms have to be in place that prevent depletion of the stem cell pool, especially when production of new neurons was boosted by injury. Our analysis of the *id1* mutant as well as the BMP gain- and loss-of-function experiments suggest exactly such a role for the BMP/Id1 signaling system. BMP/Id1 signaling promotes a non-proliferative state and thereby maintenance of the NSC pool.

After the first injury of the telencephalon, the *id1* mutants showed a stronger proliferative response than the WT siblings. When wounding was repeated, the mutants presented a significantly reduced proliferative capacity after the second round of injury. This suggests that the stem cell niche in *id1* mutants was significantly deprived of proliferation-competent NSCs after the first injury/regeneration cycle. This is reflected by a reduction of overall NSC counts after wounding. Fully in agreement with this loss of NSCs, *id1* mutants failed to repair the second lesion in contrast to WT siblings treated in the same way. *id1*, and consequently also BMP signaling, are thus required by the adult telencephalon to maintain its capacity of regenerative neurogenesis. This shift back to baseline proliferation could be facilitated by two distinct BMP/Id1- mediated mechanisms: BMP/Id1 signaling could prevent quiescent type I NSCs from entering the cell cycle or it could actively promote return of proliferation competent type II NSCs to quiescence after injury-induced rise of proliferation.

### 3.3. A Conserved Function of BMP Signaling in Stem Cell Maintenance

It is striking that the CRM2 driving *id1* expression [15] in NSCs of the telencephalon is conserved between mammals and zebrafish. Even more astonishing is that the human homologue of CRM2 mediates expression in the telencephalon of the zebrafish in a pattern identical to that of the endogenous element [15]. Moreover, the function of the id1-CRM2 depends on the integrity of conserved BMP response elements. This suggests that the basic mechanisms of BMP signaling are conserved and presumably also in operation in the two distantly related vertebrate species.

In rodents, multiple functions of BMP signaling ranging from control of proliferation, differentiation and gliogenesis to quiescence of NSCs have been reported in various contexts (see [58] for review). Fetal NSCs respond to BMPs with proliferation [59]. In contrast, blockage of BMP signaling in the dentate gyrus suggests that BMPs mediate quiescence [60] in agreement with our findings in the zebrafish. In the mouse SVZ, this control of quiescence involves locally balanced BMP levels via expression of the BMP clearance receptor LRP2 in ependymal cells. This control appears to be restricted to the SVZ as the subgranular zone of the mouse telencephalon does not express LRP2 [61]. In contrast to zebrafish NSCs, in the murine SVZ, NSCs (type B cells) express BMP2 and 4 [57] pointing at significant differences to the zebrafish. Moreover, conditionally blocking Smad4 in Type B cells of the SVZ leads to oligodendrogenesis [62]. We did not observe this shift in the zebrafish telencephalon but rather noted an increase in neurogenesis when we blocked the BMP/Id1 signaling axis. Murine type B stem cells of the SVZ show activation of Smads via phosphorylation and also ID1 is expressed at high levels in these cells [34]. Indirect evidence also points to involvement of BMPs in regulation of proliferation in the SGZ in the hippocampus of mice [58]. The BMP/Id1 regulatory module is evolutionary ancient and, like Notch signaling, it has been deployed in multiple processes during development and body homeostasis [63]. The data from studies in the mouse telencephalon suggest that the BMP/Id1 regulatory module is also employed in the control of quiescence in NSCs of the murine telencephalon, even though mice do not show a strong regenerative capacity.

### 3.4. Cooperation between BMP and Notch Pathways in the Control of Stem Cell Quiescence

Multiple lines of previous evidence suggest that Notch signaling controls proliferation of NSCs in the zebrafish telencephalon [23,27,64,65]. In particular, Notch3 has been implicated in the control of NSC quiescence [27]. Notch signaling appears to act on short-range between NSCs and activated neural precursors (type III cells) [28]. This process bears resemblance to lateral inhibition initially proposed to select the precursor cell of sensory bristles in the Drosophila cuticle [66]. Thus, the dynamics of the NSCs generate an intrinsic cue that assures their long-term maintenance [28]. Our data here suggest that intrinsic Notch signaling is not the only mechanism that safeguards maintenance of NSCs in the zebrafish telencephalon but also BMP mediated cross-talk between neurons and the stem cells.

When Notch signaling is blocked pharmaceutically, *her4.1* expression is reduced and proliferation is increased [27]. Strikingly *her4.1* expression is also affected when BMP signaling or *id1* activity is altered: Lack of *id1* function leads to a reduction of *her4.1* expression. However, inhibition of Notch signaling does not seem to influence *id1* expression [14]. Our data suggest that BMP signaling acts in parallel of and converges with Notch signaling on the expression of *her4.1*. A similar interaction of BMP and Notch signaling was recently uncovered in the control of angiogenesis in zebrafish embryos [67]. It remains to be shown, however whether the control of quiescence is mediated entirely through Her4.1 or whether other mechanisms acting in parallel exist.

So far, is unclear how Id1 and Her4.1 affect cell proliferation. During mouse neurogenesis and myogenesis, the Notch effector Hes1, related to zebrafish Her4.1, is either expressed at high levels or oscillates [68,69]. The downstream proneural gene *ascl1* is completely suppressed in quiescent stem cells in which the repressor Hes1 is expressed at high levels. Expression of the regulator Hes1 can start oscillating due to an autoinhibitory loop and, as a consequence, the expression of ascl1 becomes also oscillatory. The oscillatory state allows escape from cell-cycle arrest [68,69]. The HLH protein Id1 is a negative regulator of bHLH proteins like Her4.1 and Hes1 [32]. Moreover, Id1 binds to Her4.1 in vitro [14]. Disrupting the postulated negative feedback loop of her4.1/Hes1 would increase expression of her4.1/Hes1. In agreement, blocking BMP signaling or knocking-out id1 resulted in reduction of *her4.1* expression in the zebrafish telencephalon. It remains to be seen whether *ascl1* is indeed a key regulator of proliferation in the zebrafish telencephalon.

### 3.5. Why Are Two Pathways Necessary to Control Stem Cell Quiescence?

A major question is why are two pathways necessary to control proliferation of NSCs. Previously, it has been suggested that distinct NSCs exist with respect to marker expression and cell cycle progression along the ventricular zone of the NSCs [16,52]. A simple explanation may thus be that the distinct populations employ different mechanisms to control stem cell quiescence. However, the observation that most stem cells co-express *id1* and *her4.1* suggests that the majority of NSCs receives input from both signaling systems. The large number of NSCs expressing both *her4.1* and *id1* may be the large fraction of deep quiescent cells postulated previously [51]. In agreement, when we checked for expression of the proliferation marker PCNA, the *id1/her4.1* double-positive cells expressed PCNA less frequently than the cells expressing only *id1* or *her4.1*. It is, however, also possible that these id1+/her4.1-/PCNA+ and the id1-/her4.1+/PCNA+ cells are transition state cells that are on the way back to quiescence or in transition to neuroblasts (Type IIIa cells [16] also called activated neural progenitors (aNPs) [28].

Clearly, Notch appears to be almost sufficient to maintain a normal stem cell pool in the uninjured *id1* mutant. NSC proliferation is increased in the *id1* mutant. However, Notch maintains under these conditions a high number of NSCs,ven slightly higher than what was found in WT siblings. The stem cell population shrank only after injury in the *id1* mutant. This allocates distinct functions to Notch and BMP/Id1 signaling. While Notch3 signaling appears to have a predominantly homeostatic function during constitutive neurogenesis, BMP/Id1 comes into play when the proliferation control system is moved out of balance by injury. Under these circumstances, BMP/Id1 is necessary to prevent depletion of the stem cell pool and eventual loss of the regenerative capacity. Taken together, maintenance of stem cells by BMP/Id1 signaling is a key mechanism that underlies the remarkable ability of adult zebrafish to heal even severe injuries of the forebrain. Our data suggest that stem cells are not only maintained by niche-intrinsic cues but also via neuron/radial glial communication.

## 4. Materials and Methods

### 4.1. Zebrafish Strains and Husbandry

Experiments were performed on 6-12 month old AB wild-type (WT), *Tg(id1-CRM2:GFP)* [15], *Tg*(*her4.1:GFP*); *Tg*(*id1-CRM2:mCherry*) [54] and [15] respectively), *Tg*(*olig2:gfp*) [40], *Tg*(*hs:bmp2b*) [42], *Tg*(*hs:dnBmpr1a*) [46] and *id1^ka706/ka706^* (this study) zebrafish. Zebrafish housing and husbandry were performed following the recommendations by [70]. All animal experiments were carried out in accordance with the German protection standards and were approved by the Government of Baden-Württemberg, Regierungspräsidium Karlsruhe, Germany (Aktenzeichen 35-9185.81/G-288/18).

### 4.2. Stab Wound, Chemical Treatment and Heat-Shock of Adult Zebrafish

Stab wounding was performed as described [7,50]. In brief, after anesthesia in tricaine, we inserted a hypodermal needle directly into the right telencephalic hemisphere while the contralateral left hemisphere was kept intact and served as a control.

For the DMH1 treatment, fish were bathed in 200 mL fresh fish water containing 10 or 20 μM DMH1 diluted from a 10 mM DMH1 (Tocris, Bristol, UK) stock solution in DMSO. For LY411575 treatment, 10 mM LY411575 (Sigma Aldrich, St. Louis, MO, USA)
stock solution was diluted with fresh fish water to 6 μM or 3 uM final concentration. For the combined treatment 200 μL of a 10 mM DMH1 stock solution (final concentration 10 μM) plus 60 μL of a 10 mM LY411575 stock solution (final concentration 3 μM) were mixed with 200 mL fresh fish water. Three fish were kept in the solutions for 5 days. Every morning, fish were fed with regular adult fish food and the fish water containing the drug was changed daily.

For heat-shock, adult zebrafish were transferred to a beaker containing fresh fish water at 33–34 °C (water bath). After 30 min, the temperature was increased to 37 °C and the fish were kept for 1 hour at this temperature. Afterwards, the fish were transferred to 28.5 °C and kept for 6 more hours before being euthanized.

### 4.3. Constructs and Synthesis of Antisense DIG RNA Probes

The following antisense digoxigenin-labeled probes were used: *id1*, *her4.1*, *ascl1a* and *notch3* [71]. The *bmp2a*, *bmp7a*, *bmp7b* [72] and *bmpr2b* [73] probes were amplified by PCR from zebrafish embryonic cDNA (primers see Appendix A), then cloned into the pGEM-T easy vector (Promega). *bmp2b* and *bmp4* were kindly provided by Matthias Hammerschmidt [74] and *bmpr1ab* by Jeroen Bakkers [75]. Briefly for probe synthesis, 1 μg of each plasmid was linearized using appropriate restriction enzymes for 30 min at 37 °C. After deactivation of the restriction enzyme (see Appendix A) at 80 °C for 5 min, the plasmid was used for in-vitro RNA transcription in the presence of DIG labelling mix (Roche) and RNA Polymerase (see Appendix A) and incubated for 3 h at 37 °C. The reaction was stopped by adding 0.2 M EDTA, pH8 and purified using the ProbeQuant G50 Micro column kit (GE Healthcare). The probe was then diluted 1:1 using hybridization buffer [50] for storage at −20 °C.

### 4.4. Preparation of Adult Zebrafish Brains, In-Situ Hybridization, Immunohistochemistry, Imaging and Quantification

Brain preparation (dissection and sectioning) for ISH and immunohistochemistry were performed as described in [50].

For immunohistochemistry, primary antibodies included: chicken anti-GFP (1:1000, Aves labs, Davis, CA, USA), mouse anti-PCNA (1:500, Dako, Agilent, Santa Clara, CA, USA), rabbit anti-S100β (1:400, Dako, Agilent, Santa Clara, CA, USA), mouse anti-GS (Glutamin Synthetase) (1:1000, Millipore, Burlington, MA, USA), rabbit anti-HU (1:500, Abcam, Cambridge, UK), rabbit anti-BMP2b-Zebrafish (1:50, Anaspec, AS-55708), mouse anti-NeuroD1 (1:500, Abcam, Cambridge, UK), rabbit anti-Phospho Smad1/5/9 (1:200, Cell Signaling Technology, Danvers, MA, USA) and rabbit anti-RFP (1:500, antibodies online, Aachen, Germany). Secondary antibodies were conjugated with Alexa fluor dyes (Alexa series) and included anti-chicken Alexa 488, anti-mouse Alexa 546 and anti-rabbit Alexa 633 (1:1000, Invitrogen, Waltham, MA, USA). For ISH, the prepared DIG probes were hybridized with the brain tissue. After cutting, secondary DIG antibodies (anti-DIG-AP for chromogenic staining; anti-DIG-POD for fluorescent staining) were applied overnight. Staining took place on the next day with NBT/BCIP in the case of chromogenic staining or Tyramide Cy3 solution (Perkin Elmer, Waltham, MA, USA) for fluorescent staining. Pictures of chromogenic in situ hybridized sections were acquired with a Leica stereomicroscope MZ16 F. Immunohistochemically stained brain slices were mounted using Aqua-Poly/Mount (Cat No. 18606-20, Polysciences, Inc, Warrington, PA, USA) with coverslips (0.17mm thickness) and imaged with a laser scanning confocal microscope (Leica TCS SP5). To obtain single-cell resolution images, an HCX PL APO CS x63/1.2NA objective was used with the pinhole size set to 1-airy unit. Fluorescent images for green (GFP), red (PCNA), and infrared channesl (S100β, GS, HU, Bmp2b, NeuroD1, pSmad1/5/9, and mCherry) were acquired sequentially in 16-bit color depth with excitation/emission wavelength combinations of 488 nm/492–550 nm, 561 nm/565–605 nm, and 633 nm/650–740 nm, respectively. Pixel resolution for XY and Z planes are 0.24 and 0.50 μm, respectively. For individual brain samples, at least three transverse sections cut with a vibratome (VT1000S, Leica, Wetzlar, Germany) at different anterior-posterior levels representing anterior, posterior an intermediate telencephalic regions were imaged.

### 4.5. Real-Time Quantitative PCR

Total RNA was isolated from adult telencephala using Trizol (Life Technology, ThermoFisher Scientific, Darmstadt, Germany). First- strand cDNA was synthesized from 1 μg of total RNA with the Maxima First-Strand cDNA synthesis kit (ThermoFisher Scientific, Darmstadt, Germany) according to the manufacturer’s protocol. A StepOnePlus Real-Time PCR system (Applied Biosystems) and SYBR Green fluorescent dye (Promega, Madison, Wisconsin, USA) were used. Expression levels were normalized using β-actin. The relative levels of mRNAs were calculated using the 2^−ΔΔCT^ method. The primer sequences [76] are listed in Appendix A. Experiments were performed at least 3 times, each time with RNA pooled from 5 brains for WT or *id1^ka706/ka706^*, respectively.

### 4.6. Statistical Analysis

For the quantification of proliferating NSCs, the number of PCNA+/S100β+ Type II cells was counted in 1 μm steps of 50-µm thick z-stacks (imaged with a 63× objective). Three sections per brain from at least 3 individuals were analyzed. For quantification of *her4.1* and *ascl1a* expression in the telencephalon, sections were photographed with a stereo microscope. Staining intensity in the ventricular zones of the dorsolateral and dorsomedial parts from three individual brains was measured by ImageJ. The fold-induction was calculated for each brain as a ratio over the average of a control group. Comparisons between two data sets from quantification of proliferating NSCs, quantification of expression via ImageJ or qRT-PCR, were performed by Welch two-sample t-test. For comparisons between more than two groups, one-way ANOVA was performed to assess whether there is a difference among group means, followed by Tukey’s multiple pairwise-comparison test as a post hoc test using R software.

### 4.7. Image Analysis

Confocal brain images were opened with Fiji/ImageJ software as composite hyperstacks to manually evaluate colocalization of GFP, PCNA, S100β, GS, HU, mCherry proteins and expression of *bmp* genes and *notch3* mRNA (FISH). Cells expressing individual markers or marker combinations were counted in the dorsomedial and the dorsolateral ventricular zones in three transverse sections prepared at different anteroposterior levels of the telencephalon.

### 4.8. Generation of the id1 Knockout Allele id1^ka706^

The oligonucleotide containing the target sequence (5′-CCAAAATGAAAGTTGTGGGACCT-3′), was designed using the ZiFiT Targeter program [77]. The guide RNA was synthesized using the cloning-free guide RNA synthesis method adapted from [78] where an oligonucletide containing the T7 promoter is annealed to the gene specific target oligo. After annealing, T4 DNA Ligase was added and the mixture was incubated at 12 °C for 20 min in a thermocycler to fill up the sequence of the annealed oligonucleotides to a double stranded DNA. Afterwards, the DNA was purified by column purification (Gel and PCR clean up kit, Macherey-Nagel) and used for RNA synthesis (Megashortscript T7 Kit, Ambion).

Single-cell stage embryos were injected with 300 ng/μL Cas9 protein (GeneArtTM PlatinumTM Cas9 Nuclease; Invitrogen/Life Technologies) along with 200 ng/μL of previously synthesized guide RNA and phenol red to a final concentration of 0.05% as visual marker for injection. F0 embryos were raised to adulthood and then outcrossed with WT animals. F1 progeny with indel mutation were in-crossed, and homozygous F2 mutants were identified.

For genotyping, genomic DNA was isolated from injected embryos or fin biopsies from adult fish by the HotSHOT method [79] for determination of guide RNA efficiency as described in [80]. Genomic DNA was prepared by incubating biopsy samples in 75 μL of 25 mM KOH with 0.2 mM EDTA at 96 °C for 20 min, followed by neutralisation with 75 μL of 40 mM Tris-HCl (pH 3.8). The genomic region containing the site of mutation was PCR-amplified using the following conditions: initial denaturation step at 94 °C for 7 min; 35 cycles of 94 °C for 25 s, 56 °C for 30 s and 72 °C for 30 s; and a final elongation step at 72 °C for 10 min. For *id1*, a 451 bp amplicon encompassing the mutation site was generated using the following primers: forward 5′-CATCATCCGCAGAAGACACA-3′; reverse 5′-AACATGGTCATCTGCTCGTC-3′. The PCR product was sent for Sanger sequencing to identify the mutant alleles (Microsynth, Balgach, Swiss).

### 4.9. Single-Cell Sequence Analysis

For assessing genes co-expressed with *id1* in NSCs, we downloaded the prepared count data of 370 cells (zebrafish_neurogenesis_smartseq.h5ad in https://github.com/fabianrost84/lange_single-cell_2019, accessed on 21 September 2021). We used the Scanpy package to read this file [81]. After the quality control to remove the low-quality cells, 264 cells were used for further analysis. The number of NCSs was 76. The number of NSCs with *id1* expression was 44. The set of 44 *id1+* NSCs cells contained 22 quiescent (ccnd1-negative) cells. We calculated the Pearson’s correlation coefficient between *id1* gene expression and expression of other genes in *id1* and *ccnd1*- NSCs/RGCs by R function (cor.test).

## Figures and Tables

**Figure 1 cells-10-02794-f001:**
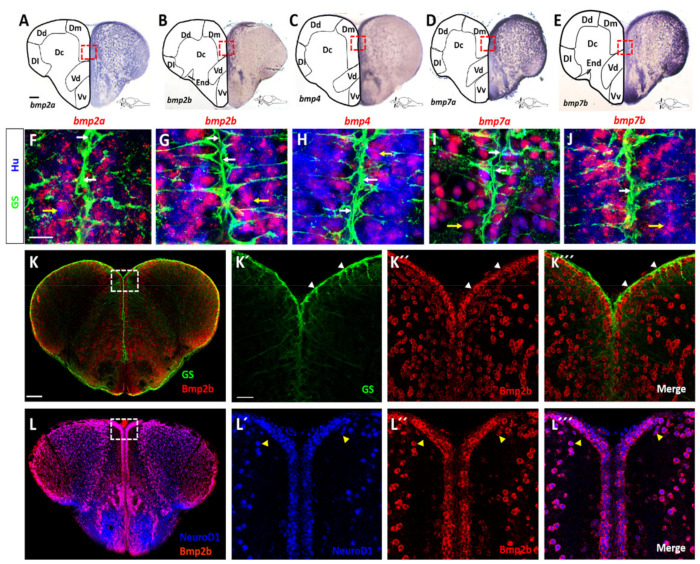
Expression of *bmp* genes in the adult zebrafish telencephalon. (**A**–**E**) ISH with probes directed against *bmp2a* (**A**), *bmp2b* (**B**), *bmp4* (**C**), *bmp7a* (**D**) and *bmp7b* (**E**) mRNA on transverse telencephalic sections. While *bmp2a* (**A**), *bmp7a* (**D**) and *bmp7b* (**E**) appear to be expressed in all telencephalic nuclei, *bmp2b* (**B**) and *bmp4* (**C**) expression is more confined to the medial region. Red rectangles in (**A**–**E**) indicate the coordinates of the region presented in (**F**–**J**), respectively. (**F**–**J**) Expression of *bmp* genes revealed by fluorescent FISH (red) on cross-sections of WT telencephala, together with double immunofluorescent staining for NSCs (glutamine synthetase [GS], green) and post-mitotic neurons (Hu, blue). The *bmp* genes are co-expressed with Hu in neurons (yellow arrows in **F**–**J**) but are not expressed in GS+ NSCs (white arrows). (**K**–**L’’’**) BMP2b antibody staining (red) along with antibodies against GS (**K**–**K’’’**, green) or NeuroD1 (**L**–**L’’’**, blue) shows that Bmp2b is expressed in NeuroD1+ neurons. White rectangles (**K**,**L**) represent the region magnified in **K’**–**K’’’** and **L’**–**L’’’**, respectively. White arrowheads show the GS+ NSCs (**K’**–**K’’’**), yellow arrowheads indicate neurons (**L’**–**L’’’**). Scale bar = 20 μm (**F**–**J, K’**–**K’’’**,**L’**–**L’’’**), 100 μm (**A**–**E**,**K**,**L**). Dc: central zone of the dorsal telencephalic area; Dl: lateral zone of the dorsal telencephalic area; Dm: medial zone of the dorsal telencephalic area; Vc: central nucleus of the ventral telencephalic area; Vd: dorsal nucleus of the ventral telencephalic nucleus; Vv: ventral nucleus of the ventral telencephalic area.

**Figure 2 cells-10-02794-f002:**
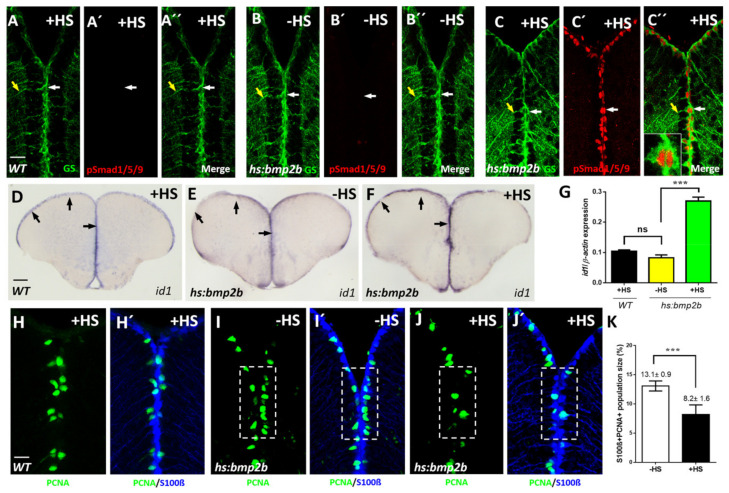
Conditional expression of *bmp2b* causes activation of BMP signaling, elevated expression of *id1* and an increased proliferation of NSCs. (**A**–**C’’**) Immunostaining with GS (green) and pSmad1/5/9 (red) antibodies on sections of WT telencephala, treated with heat-shock (**A**–**A’’**, +HS) and *Tg*(*hs:bmp2b*) telencephala without heat-shock (**B**–**B’’**, -HS) or with heat-shock (**C**–**C’’**, +HS). pSmad1/5/9 immunoreactivity was detected in GS+ NSCs exclusively in heat-shocked telencephala of the *Tg*(*hs:bmp2b*) line. (**C’’**) inset: a magnified view of a pSmad+/GS+ NSC. Note the nuclear localization of pSmad in contrast to cytoplasmic localization of GS. White arrows show individual NSCs, yellow arrows indicate the processes of the same NSCs (**A**–**C’’**). (**D**–**F**) ISH on telencephalic cross- sections indicating increased *id1* expression after heat-shock-induced bmp2b expression. Black arrows show the expression of *id1* in the ventricular zone of WT after heat-shock (**D**), *Tg*(*hs:bmp2b*) after heat-shock (**F**) and *Tg*(*hs:bmp2b*) without heat-shock (**E**). (**G**) RT-qPCR analysis of *id1* mRNA expression in WT telencephala with heat-shock (WT +HS) and *Tg*(*hs:bmp2b*) telencephala without heat-shock (-HS) and with heat-shock (+HS). (**H**–**J’**) Immunohistochemistry with antibodies against the NSC marker S100β (blue) and the proliferation marker PCNA (green) on telencephalic transverse sections from WT with heat-shock (H,H’) and *Tg*(*hs:bmp2b*) without heat-shock (**I**,**I’**) and with heat-shock (**J**,**J’**) focusing on the ventricular zone of the dorsal telencephalon. Note that the number of proliferating NSCs (PCNA+/S100β+ cells, dashed, white boxed areas) is reduced in heat-shocked telencephala of the *Tg(hs:bmp2b*) line (**J**,**J’**). (**K**) Quantification of the PCNA+/S100β+ + cells in telencephala of the *Tg*(*hs:bmp2b*) line showing a significant reduction in the heat-shocked (+HS) telencephala compared to the control group without heat-shock. Significance is indicated by asterisks: ns, not significant; *** *p* < 0.001. *n* = 3 brains (G), *n* = 15 sections (K). Scale bar = 20 μm (**A**–**C’’**,**H**–**J’**), 100 μm (**D**–**F**). HS: heat-shock.

**Figure 3 cells-10-02794-f003:**
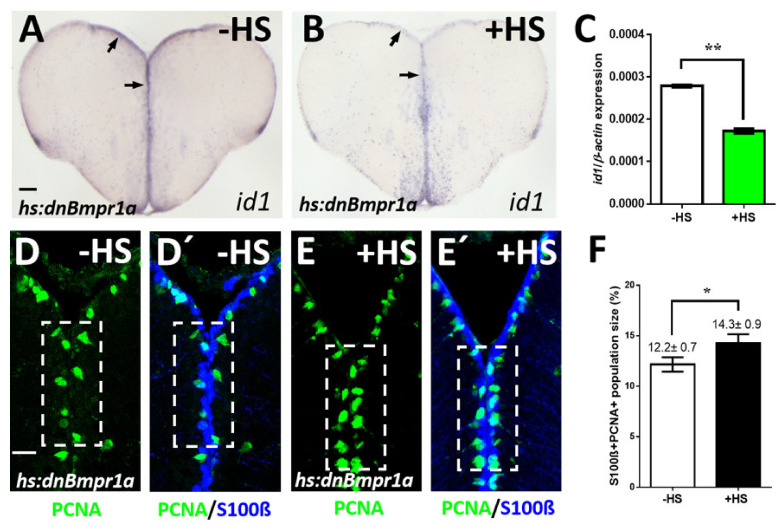
Inhibition of BMP signaling leads to a reduction of *id1* expression and an increase of proliferating NSCs. (**A**,**B**) Reduced *id1* expression after inhibition of the BMP pathway by heat-shock of *Tg*(*hs:dnBmpr1a*) animals. Black arrows point at the expression of *id1* in the ventricular zone of *Tg*(*hs:dnBmpr1a*) animals without (**A**, -HS) and with heat-shock (**B**, +HS). (**C**) RT-qPCR quantification of *id1* mRNA expression shown in A-B indicates that inhibition of BMP signaling leads to a significant reduction of *id1* expression. (**D**–**E’**) Immunohistochemistry on telencephalic transverse sections of *Tg*(*hs:dnBmpr1a*) animals for NSCs (S100β, blue) and proliferating cells (PCNA, green), focusing on the ventricular zone of the dorsal telencephalon. The number of proliferating NSCs (PCNA+/S100β+ cells) is increased after heat-shock (compare dashed white boxed area in (**D**,**D’**) to (**E**,**E’**)). (**F**) Quantification of the population size of PCNA+/S100β+ cells exhibiting a significant increase in heat-shocked (+HS) compared to non-heat-shocked (-HS) telencephala. Significance is indicated by asterisks: * 0.01 ≤ *p* < 0.05; ** *p* < 0.01. *n* = 3 brains (**C**), *n* = 15 sections (**F**). Scale bar = 20 μm (**D**–**E’**), 100 μm (**A**,**B**). HS: heat-shock.

**Figure 4 cells-10-02794-f004:**
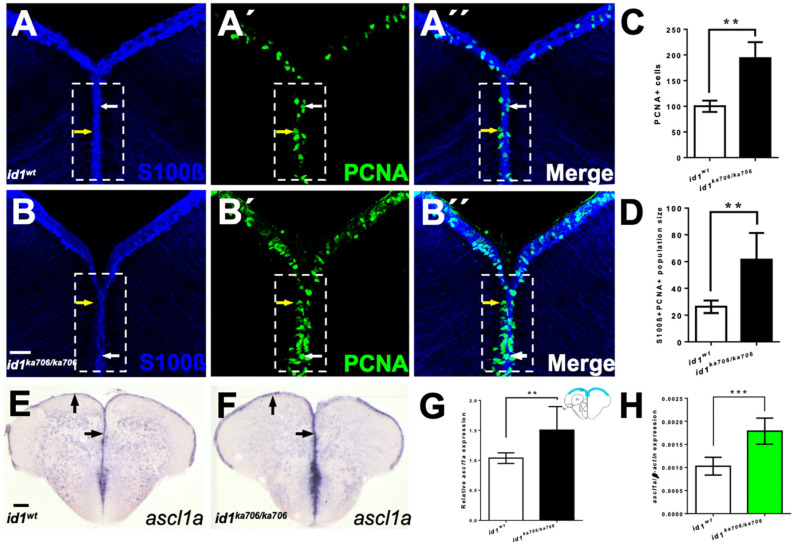
Loss of *id1* function leads to an increased number of proliferating NSCs. (**A**–**B’’**) NSCs marked by immunohistochemistry with S100β (blue) and PCNA (green) antibodies on telencephalic cross-sections from adult *id1^ka706/ka706^* and WT siblings. The number of PCNA+ cells per brain is increased in the mutant (**B’**, dashed white boxed area) compared to WT control brains (**A’**, dashed white boxed area). White arrows show PCNA+/S100β+ cells, yellow arrows show PCNA+ cells. White boxed area indicates area of quantification. (**C**) Relative population size of PCNA+ cells in WT and *id1^ka706/ka706^* brains. (**D**) Quantification of the number of proliferating NSCs (type II cells, PCNA+/S100β+ cells) in *id1^ka706/ka706^* and WT siblings. (**E**,**F**) Expression of *achaete-scute-like1a (ascl1a)* mRNA is increased in *id1^ka706/ka706^* telencephala. Black arrows show the expression of *ascl1a* in the ventricular zone of *id1^ka706/ka706^* mutants and WT siblings. (**G**) Quantification of *ascl1a* expression (scheme in the upper right-hand corner displays the quantified area in blue) in mutants and WT siblings. (**H**) RT-qPCR quantification confirms induction of *ascl1a* in *id1^ka706/ka706^*. Significance is indicated by asterisks: ** *p* < 0.01; *** *p* < 0.001. *n* = 3 brains (**A**–**D**), *n* = 15 sections (**G**), *n* = 5 telencephala (**H**). Scale bars: 20 μm (**A**–**B’’**) 100 μm (**E**,**F**).

**Figure 5 cells-10-02794-f005:**
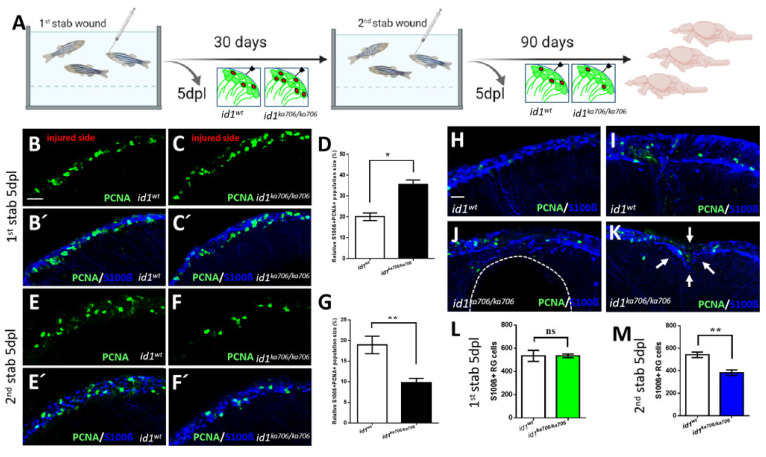
*id1* is required to maintain the stem cell pool and the regenerative capacity of the telencephalon. (**A**) Experimental layout: 6-month-old adult *id1^ka706/ka706^* fish (*n* = 25) and WT siblings (*n* = 25) were stabbed with a needle into the right hemisphere of the telencephalon. Fish (*n* = 2) of each group were sacrificed at 5 days post lesion (dpl) and the remaining fish were allowed to recover for one month before a second stab wound was inflicted. Five days after the second lesion, fish (*n* = 7) from each group were sacrificed for analysis. The remaining fish were kept for another 3 months before they were sacrificed (*n* = 16 for WT and *n* = 11 for *id1^ka706/ka706^* mutants). Note that five mutant fish showed signs of suffering after the second wounding and were sacrificed before the end of the three months recovery. (**B**–**C’**, **E**–**F’**, **H**–**K**) Double immunohistochemistry with S100β (blue) and PCNA (green) antibodies was carried out to mark proliferating NSCs on telencephalic cross-sections from *id1^ka706/ka706^* mutants and WT siblings at the different time points. (**B**–**C’**) After the first stab, *id1^ka706/ka706^* mutants showed a higher number of proliferating NSCs. (**D**) Quantification of the population size of PCNA+/S100β+ cells in relation to the total number of NSCs (S100β+ cells) in WT and *id1^ka706/ka706^* telencephalic cross-sections after the first stab wound. (**E**–**F’**) After the second injury, *id1^ka706/706^* mutants showed less proliferating NSCs (PCNA+/S100β+ cells) compared to WT siblings. (**G**) Quantification of the population size of PCNA+/S100β+ cells in relation to the total number of NSCs (S100β+ cells) in cross-sections through WT and *id1^ka706/ka706^* telencephala. (**H**–**K**) WT fish had repaired the lesion in the telencephalon 3 months after the second stab without signs of the injury (**H**) or with only mild signs of slight tissue disorganization (**I**). In contrast, *id1^ka706/ka706^* mutants showed severe tissue lesions (**J**), such as a hole in the parenchyma of the telencephalon (dashed line) or dents in the parenchyma with tissue disorganization (**K**, white arrows). (**L**,**M**) Quantification of the NSCs (S100β+ cells) in the injured side after inflicting the first (**L**) and second (**M**) stab wound in WT and *id1^ka706/ka706^* mutants showing a stepwise reduction of the number of NSCs. (Compare also with the increased number of NSCs in the uninjured mutant (Appendix A)). Significance is indicated by asterisks: ns, not significant; * 0.01≤ *p* < 0.05; ** *p* < 0.01. Scale bars: 20 μm (**B**–**C’**,**E**–**F’**,**H**–**K**).

**Figure 6 cells-10-02794-f006:**
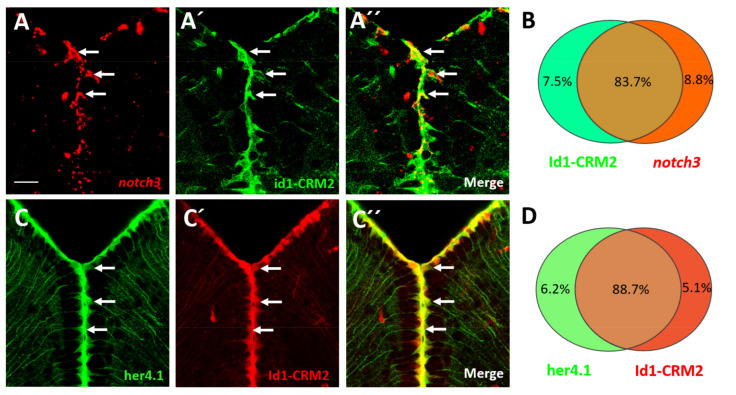
*notch3*, *her4.1* and *id1* are co-expressed in the NSCs of the telencephalon. (**A**–**A’’**) FISH against *notch3* mRNA (**A**, red) on transverse sections through the telencephalon of *Tg*(*id1-CRM2:gfp*) transgenic animals (**B**, green). **A’’** merged view of panels (**A**,**A’**). An overlapping pattern of expression for *notch3* and *Tg*(*id1-CRM2:gfp*) (green) was noted in the telencephalon (white arrows). (**B**) Summary of co-expression analyses of *Tg*(*id1-CRM2:gfp*) with *notch3*. (**C**–**C’’**) Immunostaining on cross sections of the telencephalon of *Tg*(*her4.1:gfp;id1-CRM2:mCherry*) double transgenics with antibodies against GFP (green), and mCherry (red). GFP and mCherry signals co-localize indicating that *her4.1* and id1 are co-expressed. (**D**) Summary of co-expression analyses of *Tg*(*id1-CRM2-mCherry*) with *Tg*(*her4.1:gfp*). *n* = 3 brains. Scale bar = 20 μm.

**Figure 7 cells-10-02794-f007:**
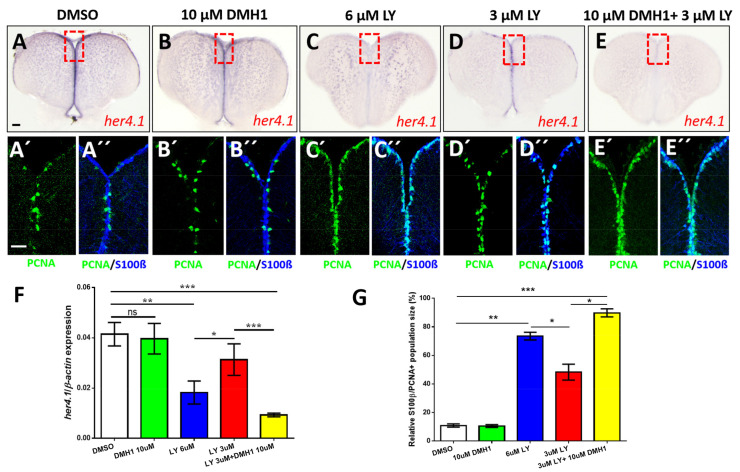
Notch and BMP signaling pathways interact to control quiescence of NSCs. (**A**–**E**) Expression of *her4.1* revealed by ISH on control telencephala (DMSO, **A**) or telencephala treated with different concentrations of DMH1 (**B**), LY411575 (LY, **C**,**D**), or a combination of LY and DMH1 (**E**). (**B**) 10 μM DMH1 does not influence strongly the expression of *her4.1*. (**C**) Notch blockage reduces *her4.1* expression along the ventricular zone. (**D**) After reduction of the concentration of LY to 3 μM, *her4.1* expression is still recognizable in the ventricular zone. (**E**) Combination of 3 μM LY with 10 μM DMH1 blocks *her4.1* expression in the ventricular zone suggesting that the two pathways interact. Red rectangles (**A**–**E**) illustrate regions of immunostaining in **A’**–**E’**. (**A’**–**E’**) Cross-sections of the pallial ventricular zone following different concentrations of DMH1 or LY treatment or a combination of both, immunostained for the NSC marker S100β (blue) and the proliferation marker PCNA (green). The proportion of PCNA+/S100β+ cells is increased in the groups treated with 6 μM LY alone (**C’**,**C’’**) or treated with a combination of 3 μM LY and 10 μM DMH1 (**E’**,**E’’**). (**F**) RT-qPCR analysis of *her4.1* mRNA expression under different conditions of drug treatment. (**G**) Quantification of the relative population size of PCNA+/S100β+ cells under different conditions. Significance is indicated by asterisks: ns, not significant; * 0.01 ≤ *p* < 0.05; ** *p* < 0.01; *** *p* < 0.001. Scale bars: 20 μm (**A’**,**A’’**,**B’**,**B’’**,**C’**,**C’’**,**D’**,**D’’**,**E’**,**E’’**), 100 μm (**A**,**B**,**C**,**D**,**E**).

## Data Availability

Not applicable.

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
