# Peer review of "Neuron-Radial Glial Cell Communication via BMP/Id1 Signaling Is Key to Long-Term Maintenance of the Regenerative Capacity of the Adult Zebrafish Telencephalon"

_cells, 2021, doi:10.3390/cells10102794_

Round 1
Reviewer 1 Report
The paper entitle “Neuron-radial glial cell communication via BMP/Id1 signaling is key to long-term maintenance of the regenerative capacity of the adult zebrafish telencephalon” by G. Zhang, L. Lübke, F. Chen, T. Beil, M. Takamiya, N. Diotel, U. Strähle and S. Rastegar is potentially interesting, however there are several concerns about this paper:
- Sometimes the reading of the present paper is not easy. The authors should try to streamline the text where it is possible, even using schemes. A graphical scheme would make reading and comprehension easier. Moreover, authors should make tighter and more logic links between the paragraphs.
- In the drafting of the manuscript the authors have not followed a homogeneous form of writing. In particular, throughout the text the authors wrote the same words in different ways using, sometime the words in extended or abbreviations for the same words (for example the abbreviations must always be the same, …..) such as in figure 1 A-E GS (glutamine synthetase) is represented in green whereas in the same figure 1 K-K’’’ is represented in red. This creates confusion in the reader.
- The legends of the figures need to be revised to make reading and comprehension easier
- The authors did not show the internal controls in the figures.
- In figure 2 the authors should show Wild-type in the different experimental conditions: -HS (heat-shock) and +HS.
- The statistics of figures 3F, 4D, 4G, and 4H need to be revised
Reviewer 2 Report
The work as a whole makes a good impression, but the summary of the article is unsuccessful, the main algorithm and idea of the work are not disclosed. The authors exaggerate somewhat when they write "The mechanisms for maintaining the pool of neural stem cells (NSCs) that underlie the regenerative capacity of the brain in adult zebrafish are not well understood." It is necessary to substantiate the relevance of the work in a softer and more complete manner and to better describe the methodology and objectives of the article in the summary section. The work used a large number of cellular and molecular biological approaches, as well as immunohistochemistry, which should be reflected in the Summary section. Moderate correction of English is required.
Author Response
Answer to Reviewer 2
The work as a whole makes a good impression, but the summary of the article is unsuccessful, the main algorithm and idea of the work are not disclosed. The authors exaggerate somewhat when they write "The mechanisms for maintaining the pool of neural stem cells (NSCs) that underlie the regenerative capacity of the brain in adult zebrafish are not well understood." It is necessary to substantiate the relevance of the work in a softer and more complete manner and to better describe the methodology and objectives of the article in the summary section. The work used a large number of cellular and molecular biological approaches, as well as immunohistochemistry, which should be reflected in the Summary section. Moderate correction of English is required.
We thank reviewer #2 for this important remark. We modified and extended the abstract of our manuscript to make it as exhaustive as possible (see our revised manuscript).
As suggested by the reviewer #2, we also changed the sentence “The mechanisms for maintaining the pool of neural stem cells (NSCs) that underlie the regenerative capacity of the brain in adult zebrafish are not well understood." To “. However, the mechanisms controlling NSC pool maintenance are not yet fully understood."
Round 2
Reviewer 1 Report
The authors, according to the referee’s suggestions, have answered that the referees had requested. The revised version of the paper entitled “Neuron-radial glial cell communication via BMP/Id1 signaling is key to long-term maintenance of the regenerative capacity of the adult zebrafish telencephalon” by G. Zhang, L. Lübke, F. Chen, T. Beil, M. Takamiya, N. Diotel, U. Strahle, S. Rastegar is now suitable for publication in Cells.